# Solvent-Free Production by Extrusion of Bio-Based Poly(glycerol-co-diacids) Sheets for the Development of Biocompatible and Electroconductive Elastomer Composites

**DOI:** 10.3390/polym14183829

**Published:** 2022-09-13

**Authors:** Shengzhi Ji, Mathilde Stricher, Frédéric Nadaud, Erwann Guenin, Christophe Egles, Frédéric Delbecq

**Affiliations:** 1Université de Technologie de Compiègne, ESCOM, TIMR (Integrated Transformations of Renewable Matter), Centre de Recherche Royallieu, CS 60 319, CEDEX, 60 203 Compiègne, France; 2Université de Technologie de Compiègne, CNRS, BMBI (Biomechanics and Bioengineering), Centre de Recherche Royallieu, CS 60 319, CEDEX, 60 203 Compiègne, France; 3UTC-SAPC, Rue du Docteur Schweitzer, CS 60319, CEDEX, 60 203 Compiègne, France

**Keywords:** bio-based elastomers, twin-screw extrusion, composite bioplastic, high processability, flexible atmospheric pH sensor

## Abstract

Faced with growing global demand for new potent, bio-based, biocompatible elastomers, the present study reports the solvent-free production of 13 pure and derived poly(glycerol-co-diacid) composite sheets exclusively using itaconic acid, sebacic acid, and 2,5-furandicarboxylic acid (FDCA) with glycerol. Herein, modified melt polycondensation and Co(II)-catalyzed polytransesterification were employed to produce all exploitable prepolymers, enabling the easy and rapid manufacturing of elastomer sheets by extrusion. Most of our samples were loaded with 4 wt% of various additives such as natural polysaccharides, synthetic polymers, and/or 25 wt% sodium chloride as porogen agents. The removal of unreacted monomers and acidic short oligomers was carried out by means of washing with NaHCO_3_ aqueous solution, and pH monitoring was conducted until efficient sheet surface neutralization. For each sheet, their surface morphologies were observed by Field-emission microscopy, and DSC was used to confirm their amorphous nature and the impact of the introduction of every additive. The chemical constitution of the materials was monitored by FTIR. Then, cytotoxicity tests were performed for six of our most promising candidates. Finally, we achieved the production of two different types of extrusion-made PGS elastomers loaded with 10 wt% PANI particulates and 4 wt% microcrystalline cellulose for adding potential electroconductivity and stability to the material, respectively. In a preliminary experiment, we showed the effectiveness of these materials as performant, time-dependent electric pH sensors when immersed in a persistent HCl atmosphere.

## 1. Introduction

Today, a great number of biodegradable poly(glycerol-co-diacids) are synthesized for various applications, especially in the biomedical fields, e.g., as cellular scaffolds to rebuild damaged tissues (e.g., cartilage, bone, nerve, or epithelial tissue), or as wound dressings [1,2]. These properties are enabled by biological recognition, enhanced cell adhesion, suitable mechanical properties, and tailored degradation kinetics. These polymers are generally formed through a typical melt polycondensation method involving the hydroxyl groups of glycerol subunits that can react with the carboxylic acid groups of diacid monomers to form successive and stable ester functions along the polymer backbone. Thus, a large variety of commercially available diacids contribute to the production of many polyesters, such as saturated or unsaturated aliphatic diacids (e.g., succinic acid, maleic acid, dodecanoic acid [3], etc.), petro-sourced terephthalic acid, and its bio-based counterpart FDCA [4].

Among these poly(glycerol-co-diacid) elastomers, poly(glycerol sebacate) (PGS) is now recognized for its remarkable cytocompatibility (in the myocardium [5], arteries [6], bone [7], and cartilage [8]) and physicochemical properties [9]. It has been studied as a surgical sealant and tissue adhesive, useful to promote the regeneration of blood vessels, retinas, and nervous tissues [10]. It can be also employed as a drug delivery system (DDS) or surgical sealant when loaded with bioactive compounds [11] or various chemicals [12]. Compared to other poly(glycerol-co-diacids), PGS expresses a shorter in vivo degradation period of up to 60 days. Furthermore, PGS composites with the addition of various fillers, such as proteins (e.g., gelatin [13], zein [14], silk [15], elastin [16]), polysaccharides (e.g., chitosan [17], cellulose [18]), synthetic polymers (e.g., polycaprolactone [19], polyvinylpyrrolidone [20], polyethylene oxide [21]), and inorganic species (e.g., TiO_2_ [22], β-tricalcium phosphate [23], Bioglass^®^ [24], SiO_2_ NPs [25], CaTiO_3_ ceramic [26], nano-hydroxyapatite [27], halloysite nanotubes [28], carbon nanotubes [29]), show enhanced performances, including better mechanical properties, hydrophilicity, adhesiveness, and biocompatibility. Many works have even reported the use of electrospinning to organize biocompatible scaffolds of PGS composites for tissue engineering or other applications in the field of biomedicine [30,31]. As a second challenge, rapid production of PGS-based flexible sensors has begun to receive considerable attention [32,33,34]. In fact, to ensure continuous monitoring of human health, for various purposes, it is important to furnish wearable electronics, such as electronic skin and neuronal implants, made of inert biocompatible polyesters [35] coupled with an electroconductive polymer such as polypyrrole [36] to avoid undesired inflammation in living organisms.

To obtain PGS scaffolds, two distinct stages are usually required: first the preparation of prepolymers, followed by thermal curing to provide a crosslinked insoluble polymer. Many studies report that a melted homogeneous solution of sebacic acid (SA) and glycerol must be stirred under an inert atmosphere in a temperature range between 120 and 140 °C for a maximum period of four days [37,38,39,40,41]. Recently, some studies have shown the influence of the atmosphere of the prepolymerization on the overall properties of the resulting PGS [42]. It has also been proven that the reaction time can be shortened down to few minutes when the mixture is heated under microwave irradiation [43,44,45]. This step affords a stable prepolymer as a brittle opaque wax that is very soluble in polar solvents with low boiling points such as tetrahydrofuran (THF), acetone, or methanol, which can be stored in a refrigerator to prevent their slow degradation. The next steps of polyester synthesis include, first, the introduction of an organic solution of the prepolymer in a mold, followed by the removal of the solvent in a vacuum before proceeding to the thermal curing. Finally, PGS is obtained as a stretchable and adhesive elastomer, sometimes displaying shape-memory ability. These properties are directly connected to the creation of internal hydrogen bond networks by the remaining free hydroxyl groups of the glycerol subunits, along with its relative high temperature sensitivity.

In this work, our strategy initially focused on the development of new and well-known pure biosourced poly(glycerol-co-diacids) and their composite materials. As illustrated in Figure 1, the initial pathway involved the preparation of each prepolymer under heating (conventional or microwave) prior to treatment in a twin-screw extruder. To the best of our knowledge, only one article has previously reported the successful use of a twin-screw extruder to produce PGS composite sheets that display shape-memory potential [23].

All elastomers and PGS composites were obtained as quite transparent or almost translucent sheets, and were studied for their physicochemical properties, pH profiles in aqueous solutions, and biocompatibility. PGS loaded with a slight amount of an electroconductive system (polystyrene sulfonate-poly (3,4-ethylenedioxythiophene) (PSSNa-PEDOT)) or a single polymer (polyaniline (PANI)) were also produced in the same manner. In summary, we intended to develop new and remarkable stimulus-sensitive electroconductive bio-based sensors with improved performance over time.

## 2. Materials and Methods

### 2.1. Materials

All chemicals were used as received, without further purification. Itaconic acid (99%) (IA), sodium chloride (NaCl), potassium persulfate (KPS or K_2_S_2_O_8_), extra-pure aniline (99.5%), cobalt(II) acetate tetrahydrate (Co(Ac)_2_, 4 H_2_O (98%), poly(sodium-p-styrenesulfonate) (PSSNa) with a molecular weight of 70,000, and 3,4-ethylenediethoxythiophene (97%) (EDOT) were purchased from Acros Organics (Geel, Belgium). Sebacic acid (98+%), microcrystalline cellulose, Pluronic^®^ F-127, furan-2,5-dicarboxylic acid (FDCA 98%), p-toluene sulfonic acid monohydrate (p-tsOH or PTSA), glycerol (reagent plus >99% GC), and lithium perchlorate (LiClO_4_ ≥ 95%) were obtained from Alfa Aesar (Tewksbury, MA, USA) and Sigma-Aldrich (St. Louis, MO, USA). The water used in all experiments was Millipore Milli-Q grade (Burlington, MA, USA).

### 2.2. Preparation of Polyester-Based Elastomers

All elastomeric extrudable materials were synthesized according to the following method: Equimolar amounts of a chosen diacid (I: itaconic or S: sebacic acids) or a mixture of two different diacids (FS: equimolar mixture of furan dicarboxylic and sebacic acids) were introduced to a 100 mL flask (total = 60.0 mmol) in the presence of 5.6 g (60.0 mmol) of glycerol. In some cases, glycerol could be loaded with 4 wt% of a selected additive and/or 25 wt% NaCl. After the addition of 0.01 g (0.05 mmol) of PTSA, the resulting dispersion was heated under an inert atmosphere in a range of temperatures from 150 to 160 °C until the mixture turned into a homogeneous solution. The stirring continued for up to three hours until a crosslinked elastomer was formed.

In the case of prepolymer synthesis assisted by microwave irradiation, all materials were introduced in the same quantities in a dedicated 30 mL vial closed with a septum. The reaction was conducted by using a microwave heating apparatus (Anton Paar Monowave 300, Anton Paar, Graz, Austria) under magnetic stirring (600 rpm) for the desired conditions of time and temperature, usually heated at 160 °C for 5–15 min.

Melt polytransesterification was carried out by means of conventional heating [46]. To obtain a robust elastomer, 5.6 g (60 mmol) of glycerol or a 4–10 wt% glycerol dispersion of one or two additives was first heated up in the presence of 1 equivalent of a dimethyl diacid ester obtained by esterification of their corresponding acids, as reported in the literature [47,48,49]. Then, 0.6 g of Co(Ac)_2_, 4 H_2_O was added to the melted solution, and the solvent-free mixture was stirred in neat condition at 160 °C for a minimum of three hours under an inert gas stream. The reaction was stopped when the agitation started to become difficult, leading to the formation of a crosslinked and colored elastomer ready for the extrusion process.

The progress of these above procedures was monitored by nuclear magnetic resonance (^1^H NMR) spectra recorded on a Bruker Avance III 400 spectrometer (Bruker, Billerica, MA, USA), using DMSO-d_6_ as solvent. Only prepolymers of Poly(glycerol-co-itaconate) (PGI) and Poly(glycerol-co-2,5-furandicarboxylate-co-sebacate) (PGFS) were studied, and their chemical shifts (δ) are quoted in parts per million (ppm). The coupling constants are quoted in Hz. For the PGS prepolymers, their ^1^H NMR spectra corroborate those found in the literature [50].

### 2.3. Extrusion of PGI, PGS, PGFS, and PGS Composites

A lab-scaled HAAKE MiniLab 3 (Thermo Scientific, Waltham, MA, USA) twin-screw extruder was employed to obtain molded elastomer sheets of 80 mm length, 20 mm width, and 1.5 mm thickness. The samples were obtained through a recirculation of the prepolymer inside the apparatus. The screw length of the extruder was 110 mm, and for each experiment, 4–5 g of a chosen prepolymer was extruded as follows: The extruder chamber was filled partwise with small fractions cut from each prepolymer as a starting material. Controlling the torque and the inner pressure, the feeding of the extruder took 5 min on average, and the chamber temperature set points ranged from 50 °C to the final temperature of 180 °C. The screw speed was also set to a maximum value of 200 rpm, and the extrusion took about 20 min to be achieved.

### 2.4. Removal of Unreacted Monomers and Surface pH Monitoring

In order to remove unreacted acidic monomers and short oligomers, and also to neutralize the surface of the bioplastic sheets, each sample was introduced to 100 mL of a saturated sodium hydrogen carbonate (NaHCO_3_) aqueous solution for a period of 24–48 h. Then, the samples were transferred into deionized water to ensure the constant pH neutrality of the surrounding medium, and dried in vacuum prior to the subsequent analysis.

### 2.5. Fourier-Transform Infrared (FTIR)

The FTIR (ATR) spectra of the pure and composite polyesters were recorded on a Nicolet iS5 FTIR (Thermo Scientific) spectrometer in the range of 400 to 4000 cm^−1^, operated at 4 cm^−1^ resolution and equipped with an ATR plate including a ZnSe crystal.

### 2.6. Field-Emission Scanning Electron Microscopy (FE-SEM)

The observations of the elastomer surfaces were conducted using a Quanta FEG 250 (FEI, Hillsboro, OR, USA) microscope with a small fragment of each elastomer deposited on a carbon-coated Pd-Pt grid and dried for 1 h under reduced pressure. The SEM micrographs were acquired in secondary electron mode, and were obtained under a low vacuum, 15 kV accelerating voltage, and 10 mm working distance.

### 2.7. Differential Scanning Calorimetry (DSC)

Experiments were conducted on a DSC 8 MC (Mettler Toledo, Giessen, Germany) using aluminum pans. Scans were conducted under nitrogen at a heating rate of 10 °C/min, in a temperature range varying between 0 and 250 °C.

### 2.8. Production of Electroconductive Polymer Glycerol Solutions

For PSSNa-PEDOT, 0.32 g of PSSNa and 0.48 g (3.36 mmol) of EDOT were added successively to 5.6 g (60.0 mmol) of glycerol. Under an inert atmosphere, a slight amount of KPS (0.04 g; 0.25 mmol) was added to the clear solution, and the resulting mixture was stirred for 1 h at 130 °C until the mixture turned dark blue. The procedure was scalable and employed without further purification in the preparation of the prepolymers by polycondensation, as described in Section 2.2. Then, the resulting composite material was loaded with PSSNa-PEDOT at a concentration near to 4 wt%.

For the 4 wt% PANI glycerol solution, we employed a modified procedure [51]. In 5.6 g of glycerol, 0.31 g (3.38 mmol) of aniline dispersion was vigorously stirred during the addition of 0.04 g (0.15 mmol) of KPS. Then, the mixture was also stirred at 130 °C for 4 h to afford a dark brown solution of PANI particles. A 10 wt% PANI solution was obtained by the same manner. On the other hand, a 10 wt% emeraldine PANI glycerol dispersion with better electroconductive performance was prepared as follows: 2.5 g (9.25 mmol) of KPS dissolved in 100 mL of distilled water was added dropwise to 100 mL of 1.0 M HCl aqueous solution containing 2.5 mL (2.55 g; 27.4 mmol) of fresh aniline, under stirring at room temperature. On the next day, the dark blue/green dispersion was filtered to remove short, water-soluble oligomers of PANI, washed with clean water, and the desired emeraldine PANI was recovered as a green solid. The material was then dried in vacuum, and 0.56 g of the solid was introduced to 5.1 g of glycerol to obtain a 10 wt% PANI dispersion that could provide a solution at temperatures above 150 °C, especially in the presence of a diacid monomer.

### 2.9. Electroconductivity and pH Sensitivity Study of PANI Conductive Polymer Blends

Two kinds of PGS–10 wt% PANI composite sheets (PGS8 and 9) were successively tested for their conductivity and pH sensitivity according this two-step procedure: First, a selected polymer sheet was connected by both of its extremities in series between a generator and a potentiometer (Leybold^®^, Hurth, Germany). Rapidly, the elastomer band was introduced to a beaker filled with 20 mL of 37% HCl aqueous solution and closed with aluminum foil. During the measurement, all direct contact with the acidic solution was strictly avoided. The generator delivered a current at a voltage between 15 and 20 volts, and the evolution of the current circulating in the plastic sheet was read directly on the screen of the potentiometer as a function of time.

In parallel, to corroborate our results, cyclic voltammetry (CV) measurements were also conducted after having performed the above pH sensitivity study using a three-electrode system on Metrohm Autolab PGSTAT101 (Metrohm Autolab, Utrecht, Netherlands) monitored using the NOVA software (Chongqing, China). This procedure involved a platinum wire as the counter electrode, a saturated calomel electrode (SCE) as the reference electrode, and PANI elastomer blend PGS8 or 9 as the working electrode. A 0.1 M LiClO_4_ aqueous solution was used as the electrolyte, and the CV curves were recorded at scan rates of 100 mV·s^−1^ for a variation of potential found between −2.0 and +2.0 V.

## 3. Results and Discussion

### 3.1. Optimization of Prepolymer Synthesis and Extrusion Process 

Prepolymers were prepared by mixing diacids with glycerol in the presence of a catalytic amount of paratoluene sulfonic acid (p-TSA). Adding a slight amount of PTSA to the mixture of monomers increased the polycondensation rate and, under conventional heating at 150–160 °C, the reaction time was reduced to a few hours, without apparent degradation of the material. The polycondensation procedure led to the formation of linear or branched polymer backbones (Figure 2) due to the branching unit nature (B3) of glycerol compared to the linear structure (A2) of the diacids. In the A2 + B3 system, usually performed in bulk conditions for greater conversion of all monomers engaged in the reaction, the thermoset behavior of the material evolved closely with the increasing number of the ester functions.

Several prepolymers were produced under conventional heating. For pre-PGI and pre-PGS, glycerol and diacid monomers were mixed at a ratio of 1:1; in the case of pre-PGFS, the ratio was set up at 1:0.5:0.5. To form prepolymers with additives, prior to the polycondensation process, 4 wt% of a selected polysaccharide (e.g., flaxseed gum, micro/nanocellulose, almond gum) in a defined volume of glycerol (results not shown here) was added to the formulation. At this concentration, each biopolymer was almost soluble in the hot liquid.

According the SEM observations of the surfaces’ morphologies, each additive showed a particular influence on the reticulated networks of the materials, which were often covered with holes and fractures (Appendix A, see Appendix A).

Then, the same process was repeated for inorganic species such as graphene oxide (GO) or NaCl; the solids remained dispersed in the glycerol itself prior to the prepolymer synthesis. The synthesis of the electroconductive additives PSSNa:PEDOT and PANI is described in the Materials and Methods, and the preparation of the prepolymers was performed in the same manner as that of the other additives. The different prepared samples are listed in Table 1.

For all of the prepolymers, the control of the temperature was necessary. For example, for pre-PGI1, the mixture was initially heated at 160 °C, and the reaction required 4 h to produce the desired prepolymer. For the pre-PGI2 prepolymer, by reducing the temperature to 150 °C, the reaction rate was reduced, requiring 10 h on average to form a material similar to pre-PGI1. Compared to SA, itaconic acid (IA) could provide the elastomer with some additional properties, especially due to the existence of a double bond suitable to create an anchor point that could allow post-functionalizations such as grafting of biomolecules. Secondly, to date, there is no example in the literature of poly(glycerol-co-diacids) involving IA as a co-monomer subunit, although other types of polyesters have been prepared [52,53,54].

For pre-PGS 1-6, when 4 wt% of an external component was added to the bulk reaction, each polycondensation needed a minimum of 8 h to produce a robust prepolymer. Moreover, the pre-PGS7 prepolymer loaded with 4 wt% nitrogen-rich PANI could generate interactions between nitrogen atoms and the carboxylic acid functions of the monomers, reducing the total amount of protons available for the polycondensation mechanism.

A series of terpolymers or poly(glycerol-2,5-furan dicarboxylate-co-sebacate) (PGFS) was also developed. PGFS is an elastomer made of glycerol, SA, and FDCA mixed at various ratios. By gradually replacing a percentage of SA with its FDCA counterpart in the polymer backbone, new and interesting material properties appeared, evolving from a soft and weak elastomer to a more elastic and sticky material (Appendix A, see Appendix A). For our extrusion procedure, we focused on prepolymers at a ratio of 1:05:05 (pre-PGFS1), forming a very sticky and white paste. A second prepolymer—pre-PGFS2, loaded with 25 wt% NaCl—was also produced.

As has already been proven, that reaction time could be shortened down to a few minutes when microwave irradiation was used as a heating source [42,44,45]; the same prepolymer preparations were thus conducted under microwave irradiation. The production of poly(glycerol itaconate) (pre-PGI) prepolymers was carried out under microwave heating using an equimolar mixture of glycerol and IA stirred at 160 °C in the presence of traces of PTSA. Even by shortening the reaction time from 15 to 5 min, the transparent, originally liquid prepolymer quickly became rigid, rubbery, and difficult to recover from the glassware, and could not be used for the next step of the extrusion. On the other hand, using the same above conditions, with a slight amount of PTSA, under microwave irradiation, PGS prepolymers (pre-PGS) were formed as the typical white wax currently described in the literature (Appendix A, see Appendix A).

### 3.2. Extrusion of PGI, PGS, PGFS, and PGS Composites

In the next stage of our study, all previous prepolymers produced by conventional heating and reported in Table 1 were extruded. A lab-scaled twin-screw extruder was employed to obtain molded elastomer sheets by adding the prepolymers to the chamber. The chamber temperature increased from 50 to 180 °C. The mixture was recirculated in the extruder chamber, and stable sheets were isolated. The results are shown in Figure 3:

As depicted in Figure 3, depending on the nature of each additive, clear variations in the sheet translucency and color were noticed. For PGI1 and 2, both materials were not sticky, but only PGI1 was almost translucent. However, except for PGS3, which contained 4 wt% microcrystalline cellulose, all elastomers were able to adhere to glassware or other types of smooth surface. Prepolymers of PGS heated at 150 °C during their synthesis generated slightly colored and translucent materials. Some degree of translucency was observed with Pluronic^®^ F-127 for PGS2 and PGFS2. The addition of NaCl seriously affected the structure of the isolated sheets. Conversely, once inert additives were replaced with 4 wt% of very colorful components such as the electroconductive polymers PEDOT or PANI, the resulting samples of PGS6 and 7 turned into deep blue and dark brown sheets, respectively. The color was uniformly dispersed in the network.

On the other hand, when using prepolymers prepared by microwave irradiation, despite our efforts, the pre-PGS wax only melted in the recirculating extruder chamber, and never produced the desired PGS sheets after 2 h of circulation time.

### 3.3. FE-SEM Observations and Spectroscopic Analysis

As depicted in Figure 4, each sheet showed a specific and different surface based on the nature and amount of the additives included over the entire thickness of their networks. PGI1’s surface was not regular, showing a lot of deformations and short fractures. The result was similar for PGI2. PGS1 showed a sharper microrelief. PGS2 demonstrated small and visible geometric cavities purposely generated by the addition of NaCl as a porogen agent. For PGS3 and PGS4, the addition of any kind of cellulose had a good impact on the surface homogeneity, due to the additional ester functions created between the hydroxyl groups of the polysaccharides and the carboxylic acid groups of SA [55]. Interestingly, for PGS5, the same phenomenon occurred. By adding 4 wt% Pluronic^®^ to the prepolymer, even with an enlarged view (Appendix A, see Appendix A), neither fractures nor defects were detected on the surface of the elastomer. According to its polyether nature, it was also possible to stabilize the inner polymer matrix by means of a successive and strong hydrogen bond network promoted by the Pluronic’s ether functions and accessible hydroxyl groups.

PGS6 and 7, with the incorporation of a slight amount of PSSNa/PEDOT or PANI, respectively, showed a clear impact on their surface morphologies. PGS6 was layered, displaying some parallel arrays, while PGS7 showed a much more wrinkled and sinuous aspect. In the zoomed view shown in Figure 5a, the PGS7 sheet surface displays deep cavities that should be interconnected without the use of a leachable porogen agent such as NaCl. The shape of the surface is influenced by the multiple evident ionic interactions permitted by the nitrogen-rich nature of PANI and the carboxylic acid functions of the oligomers contained within the elastomer.

In Figure 6, PGFS1 shows a smooth surface. As we suspected, the extrusion of PGFS2 loaded with NaCl as a porogen failed to produce a porous surface like that of PGS2, due to its excessive plasticity.

From a molecular point of view, the FTIR analyses provided in Figure 7 and Figure 8 constitute an important source of information about the overall integrity and crosslinking mechanisms involved in the synthetized materials—especially following the extrusion treatment. In Figure 7, both PGI1 and 2 show four characteristic bands at 3300, 2900, 1740, and 1150 cm^−1^, representative of –OH stretching, alkane or CH_2_ moieties, C=O stretching, and C–O of ester function, respectively. Two additional signals corresponding to CH=CH_2_ stretching appeared at 819 and 1638 cm^−1^. Furthermore, a medium band located at 3400 cm^−1^ suggested moderate hydrogen band networks between free hydroxyl groups of the materials. PGFS1 and 2 demonstrated the persistence of bands from carboxylic groups currently observed at 1410 cm^−1^—a situation that could explain the subsequent strong acidic nature of the material. In these spectra, three bands found at 764, 832, and 963 cm^−1^ were observed, corresponding to mono- and disubstituted furan rings. In addition, the ^1^H NMR chart of the prepolymers of pre-PGFS showed too much variety of signals found between 7 and 8 ppm, which could represent the imperfect involvement of FDCA in the polymer backbone [46].

In Figure 8, PGS1–7 demonstrate two distinct bands attributed to alkane groups and located in the range between 2850 and 2920 cm^−1^. Except for PGS6, all ester functions appeared as a single vibration band. Compared to these bands, the weakness of the –OH band is a consequence of a high esterification degree (ED), which must have been reinforced during the extrusion process. Moreover, at 4 wt% concentration of all additives, the FTIR detector was not sensitive enough to detect their presence through their characteristic peaks. We must note that the extrusion process is perfectly repeatable, as the FTIR signals emitted by different samples extruded from the same prepolymers were perfectly identical.

### 3.4. Calorimetry and pH Monitoring of the Materials

All results of the DSC analysis of each elastomer sheet produced by extrusion are reported in Table 2. For the most representative samples, the results of the measurements were all reported in the Appendix A (see Appendix A).

For the PGI1 sheet, due to the higher temperature of 160 °C employed for the preparation of its corresponding prepolymer precursor, only a crystallization exotherm was recorded at 177 °C, corroborating its supposed advanced level of reticulation compared to PGI2. Thus, PGI2 is clearly more amorphous than PGI1, with two successive melting bands of Tm found at 55 and 83 °C. More specifically, Tm1 corresponds to the departure of entrapped molecules of water, while Tm2 is representative of the internal hydrogen bond networks inside the elastomer sheets. The lower Ttc of 136 °C could also confirm the lower ED, explained by the lower temperature applied during the prepolymer preparation, despite the superior temperature of 180 °C that is usually necessary for generating the elastomer sheets. For PGFS1, only a superior Ttc of 190 °C was detected, and its value was seriously reduced to 56 °C due to the presence of a great amount of NaCl inside the network of PGFS2. We can assume that when NaCl is added to a solid polymer, the salt can act as a nucleating agent, reducing the normal value of Ttc. On the other hand, PGS1 gave typical values of both Tm and Ttc, found at 37 and 141 °C, respectively. In PGS2, the presence of the NaCl once again had the same effect already observed for PGFS2, reducing the initial value of Ttc to 124 °C. The addition of 4 wt% microcellulose in PGS3 allowed the temperature of both events to be reduced by 10 °C. With nanocellulose, we did not observe exactly the same incidence, and compared to PGS1 another Tm was detected at 74 °C. PGS4, 5, and 6 showed three distinct events, including two Tms and one Ttc. All materials showed a principal value of Tm varying around 38 °C. For PGS5 and 6, Tm values lower than 10 °C could correspond to the cleavage of weak interactions between the polymeric additives and the polymer covalent network. Furthermore, the introduction of a slight amount of Pluronic or PSSNa/PEDOT to the elastomer delayed the reticulation phenomenon of the material at higher temperatures. For PGS7, the presence of a nitrogen-rich polymer prevented possible undesired degradation at working temperatures between 140 and 250 °C.

### 3.5. Alternative Polytransesterification for the Production of Non-Acidic Sheets

For the purpose of biological studies, sheets with neutral surfaces were needed. Therefore, the materials were neutralized while removing entrapped and unreacted acidic monomers by immersing each elastomer sheet in a saturated NaHCO_3_ water bath. Unfortunately, PGI1 and 2 were subject to rapid degradation. On the other hand, PGFS1 and 2 continued to express corrosive behavior by releasing acidic monomers or shorter water-soluble oligomers, despite repeated washing. PGS1–7 released their remaining acidic fragments after a period of 8 days when immersed in pure water. In all cases, no swelling phenomena were observed.

Thus, to solve the degradation problems encountered with PGI1 and 2 during the monomer extraction in a basic medium, and to suppress the excessive acidity of PGFS1 and 2, another method was developed to produce these materials. Our group has already developed a mild method for producing poly(glycerol-co-diacids) directly from a mixture of glycerol and a dimethyl ester by melt polytransesterification in the presence of a cobalt(II) salt [46]. Therefore, exploitable prepolymers decorated only with ester functions were prepared, as depicted in the Figure 9. Simultaneously, we intended to evaluate the real impact of the introduction of 4 wt% PANI in the PGS sheets—especially on their surface morphology.

Thus, three further sheets were produced from dimethyl itaconate, 2,5-furandicarboxylate, and sebacate, listed as PGI-Co and PGS7-Co when containing 4 wt% PANI, and PGFS-Co when loaded with 25 wt% NaCl. All prepolymers were extruded using the unchanged previous conditions of a final temperature of 180 °C and a screw rotation speed of 200 rpm. As illustrated in Figure 10, except for PGFS-Co, two sturdy and purple sheets were easily recovered from the apparatus and immersed in deionized water. A rapid discoloration of the sheets was observed, as depicted in Figure 10.

For PGI-Co and PGS7-Co, because the pH of their washing water remained neutral, PGI-Co was subjected to some swelling despite the supposed pronounced hydrophobic character of the ester-functionalized sheet. For PGFS-Co, the material remained strongly acidic. The SEM observations shown in Figure 10 indicate the good homogeneity of the surfaces but, once again, the strong sticky character of PGFS prevented the creation of holes in the network. To confirm the potential of our polytransesterification process, the FTIR results were compared with those of the elastomer counterparts obtained via melt polycondensation (Figure 10b), and showed no clear differences in their global composition. These materials seemed to be suitable for use in the next stage, i.e., the cytotoxicity tests. For further details of the procedure, see the ESI results.

As shown in Figure 10d, only elastomers produced by the classic melt polycondensation method expressed good cytocompatibility. Details of the cytotoxicity tests can be found in the ESI results. It is easy to imagine that cobalt traces could demonstrate toxicity against L929 cell lines under the ISO10993 standard protocol. On the other hand, the neutral polymer sheets of PGS3, 6, and 7, as expected, showed better cytocompatibility. The presence of 4 wt% of an electroconductive polymer caused moderate cytotoxicity and a bad impact on the general biocompatibility of these materials [56].

### 3.6. Study of a PGS Sheet Doped with 10 wt% PANI as a pH Sensor

Compared to PGS6, PGS7 was better tolerated, and we decided to focus only on PGS doped with PANI to develop an effective pH sensor sheet obtained through active extrusion.

In view of its “insulator character”, the charge of PANI in the PGS sheet was enhanced for further investigations. As shown in Figure 11, the material PGS8—containing both 4 wt% microcellulose and 10 wt% PANI—was first produced as a dark brown, flexible, sticky sheet. When the sheet was immersed in 50 mL of deionized water, the pH of the liquid never went below 7. On the basis of preliminary electric tests, the electrical conductivity of the sheet gave an estimated value of 1.66.10^−8^ S·cm^−1^.

In the FTIR spectrum of PGS8, compared to the charts of PGS7, the characteristic bands of the emeraldine PANI corresponding to C–H bonding were clearly observed at 1034 and 821 cm^−1^, as well as bands corresponding to C-C benzenoid stretching at 1504 cm^−1^ and, finally, to C=N quinoid stretching at 1599 cm^−1^ [57]. Because the electrical conductivity of PGS8 was close to the limit of the operating zone area [32], another pathway was employed to produce a second type of PGS loaded with emeraldine PANI, as detailed in Figure 12 [57].

A final material containing 10 wt% emeraldine PANI was produced, listed as PGS9. Herein, a desired amount of the dried green solid in its protonated form was simply dispersed in glycerol prior to the melt polycondensation and, finally, a second type of elastomer candidate was also generated by extrusion. Unfortunately, this time, when immersed in a water bath, the pH value decreased to below 4. As with PGS8, similar characteristic signals were observed in its FTIR spectrum (Appendix A, see Appendix A). During the preliminary electroconduction test, no improvement of the electrical conductivity was observed for PGS9. In parallel, to measure the impact of both the amount and nature of PANI introduced, images of FE-SEM observation were also taken for a deep comprehensive study of the phenomena reported below, as shown in Figure 13.

In these images, it is possible to see four distinct and different morphologies of the surface, according the difference in polarity between PGS7 and its PGS7-Co counterparts, for the same 4 wt% concentration of loaded PANI. However, the most important difference lies in the fact that once the PANI concentration reached a value of 10 wt%, for both PGS8 and 9, their surfaces became more flat, with the appearance of twisted bands aligned along their lengths with the PANI fibers, which could improve the electroconductivity of the material.

Some recent papers encouraged us to determine the electroconductivity of our elastomer sheets when doped with a strong acid [58,59]. Nevertheless, we carried out further investigation by measuring the response of our pH sensor candidates when placed in a closed and acidic atmosphere of HCl. These results are shown in Figure 14.

For PGS8, the circulating voltage started at 750 mV, and 60 min later, the voltage reached a maximum value of 2500 mV. The discharge was very rapid after the return of the sheet to a normal atmosphere, and took only 10 min to approach a final value of 1200 mV. Surprisingly, PGS9 needed more time to increase the voltage from 1130 to 3750 mV, and the inverse decrease time in the normal atmosphere followed the same pattern as for PGS8, with a period of 25 min to fall from 4110 mV back to its initial voltage value. During our experiments, we also noticed that direct finger-skin contact could increase the value of the measured voltage by 10 mV. In fact, emeraldine PANI is known as a material that can be doped via electrochemical oxidation or direct protonation. The increase in H+ along the macromolecule backbone changes its spin pairing, and produces a new global state without variation in the number of electrons available. After performing the tests, the sheets were dried in a vacuum, and we controlled the electroconductivity of both PGS8 and 9 sheets via cyclic voltammetry (Appendix A, see Appendix A). Their relative behaviors were also demonstrated on their cyclic voltammograms (CV) conducted in aqueous 0.1 M LiClO_4_ solution. Indeed, PGS9 did not show any powerful resistance that could correspond with the rapid increase of voltage, while for PGS8 between −2000 and 400 mV a reversible redox phenomenon occurred due to the possible partial oxidation state of PANI contained in the PGS8.

## 4. Conclusions

In this work, PGS was shown to be the best bio-based elastomer in the field of tissue engineering and biomedicine, with a strong potential for producing hybrid and composite materials—especially compared to PGI. As an A2 monomer subunit, compared to SA, FDCA did not integer properly in the terpolymer backbone of PGFS—especially when employed in a typical melt polycondensation. An alternative polytransesterification method catalyzed by a Co(II) salt did not solve these problems. However, starting from an improved method for producing biocompatible PGS composite elastomers using recirculating extrusion, this study achieved the rapid development of two different kinds of performant PGS sheets blended with 10 wt% PANI as time-dependent electric pH sensors. In the near future, our intention is to exploit our concept by incorporating small fractions of these materials in more complicated devices, such as microelectromechanical systems (MEMSs) [60] or supercapacitors [61].

## Figures and Tables

**Figure 1 polymers-14-03829-f001:**
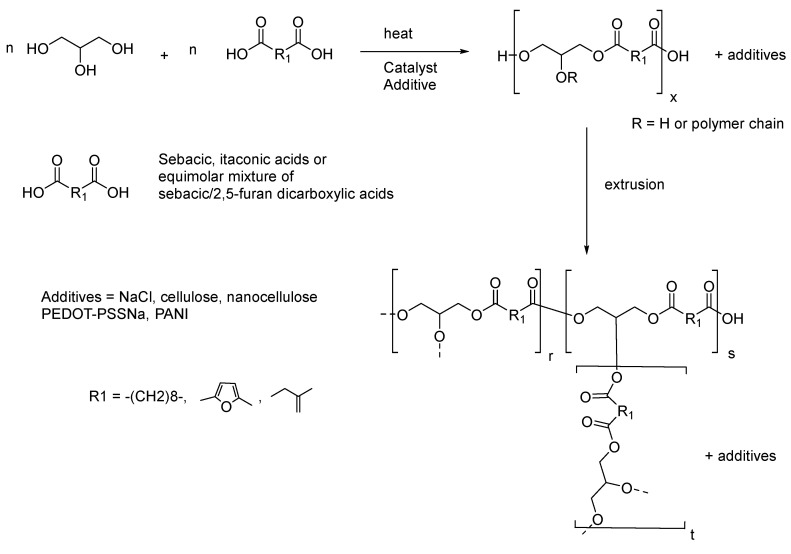
General procedure for the production of all composite elastomers.

**Figure 2 polymers-14-03829-f002:**
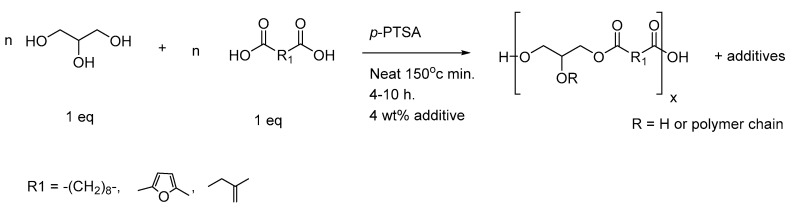
Modified synthetic pathway for producing poly(glycerol-co-diacids) composites (R = hydrogen or polymer chain).

**Figure 3 polymers-14-03829-f003:**
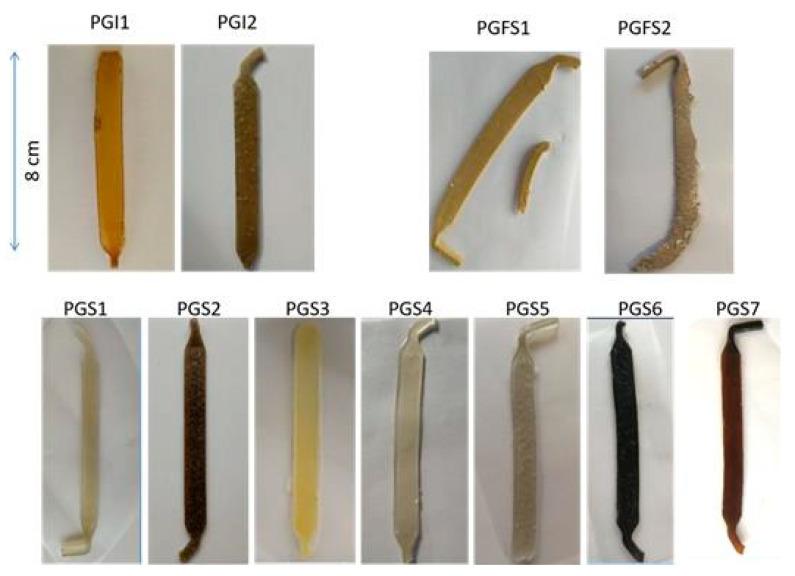
Photographs of elastomer sheets obtained by extrusion of their prepolymers when driven by the twin screws at a speed of 200 rpm and heated at 180 °C.

**Figure 4 polymers-14-03829-f004:**
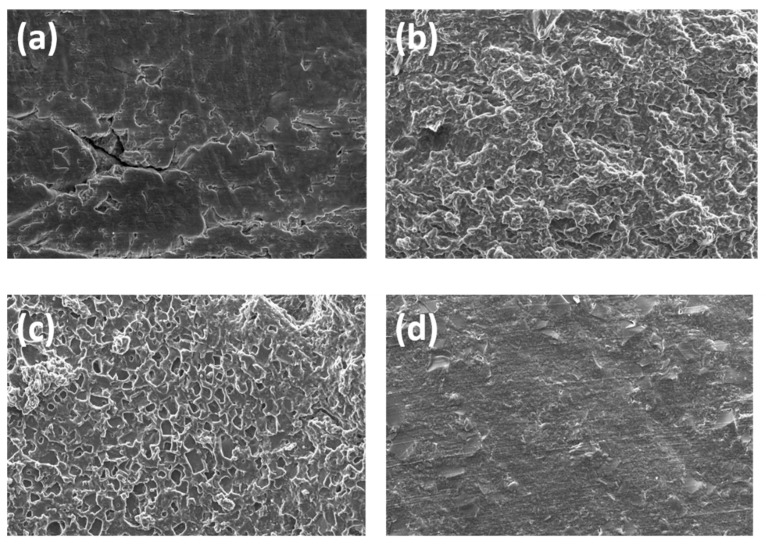
Photographs of elastomer sheets’ surfaces by FE-SEM observations: (**a**) PGI1; (**b**) PGS1; (**c**) PGS2; (**d**) PGS3. Scale bars = 400 µm.

**Figure 5 polymers-14-03829-f005:**
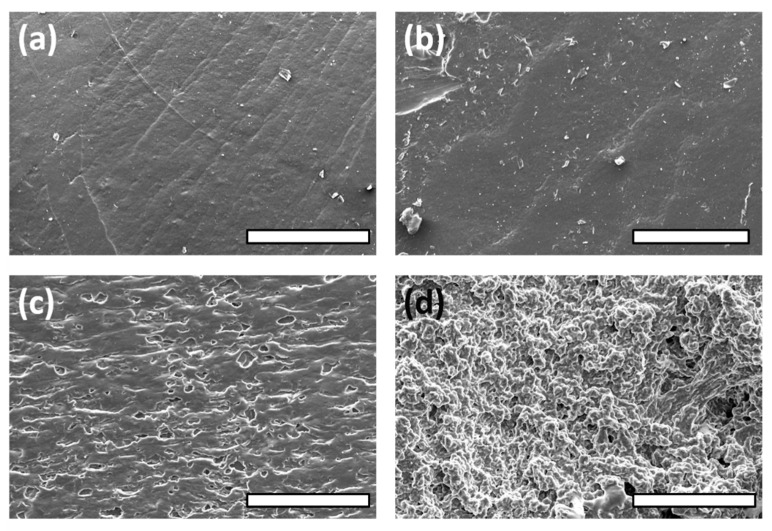
Photographs of elastomer sheets’ surfaces by FE-SEM observations: (**a**) PGS4; (**b**) PGS5; (**c**) PGS6; (**d**) PGS7. Scale bars = 400 µm.

**Figure 6 polymers-14-03829-f006:**
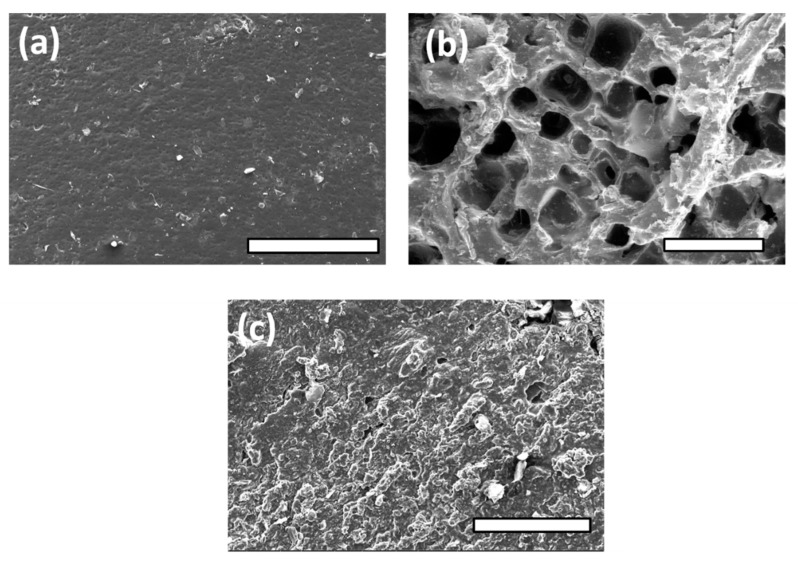
Photographs of elastomer sheets’ surfaces by FE-SEM observations: (**a**) PGFS1; (**b**) PGFS2 (prepolymer after leaching of NaCl in deionized water); (**c**) PGFS2 (after extrusion process). Scale bars = 400 µm for (**a**,**c**), and 1 mm for (**b**).

**Figure 7 polymers-14-03829-f007:**
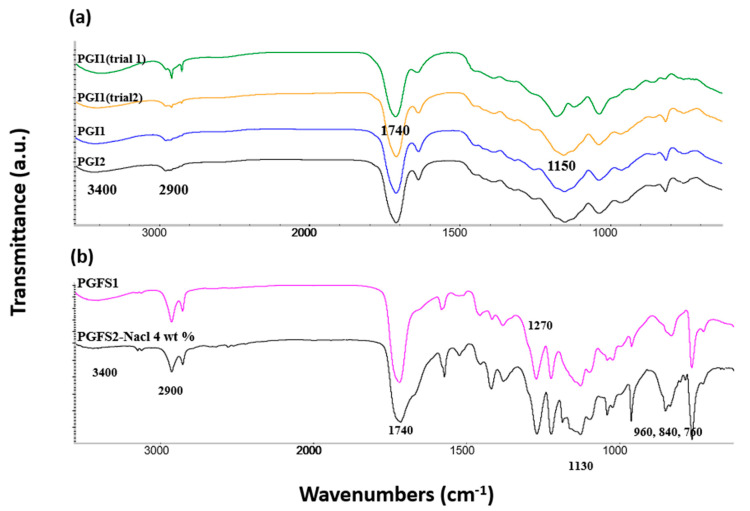
Infrared spectra of elastomer sheets: (**a**) for PGI 1 and 2 (and from prepolymers obtained under microwave irradiation); (**b**) for PGFS1 and 2.

**Figure 8 polymers-14-03829-f008:**
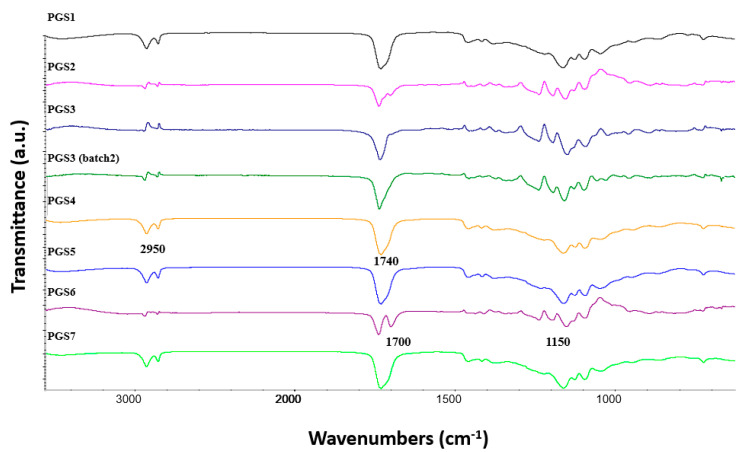
Infrared spectra of elastomer sheets from PGS1 to PGS7.

**Figure 9 polymers-14-03829-f009:**
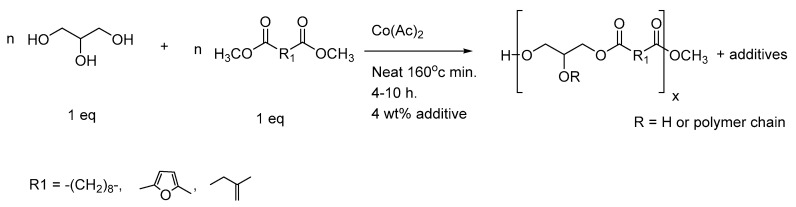
Alternative pathway to access to ester-functionalized prepolymers.

**Figure 10 polymers-14-03829-f010:**
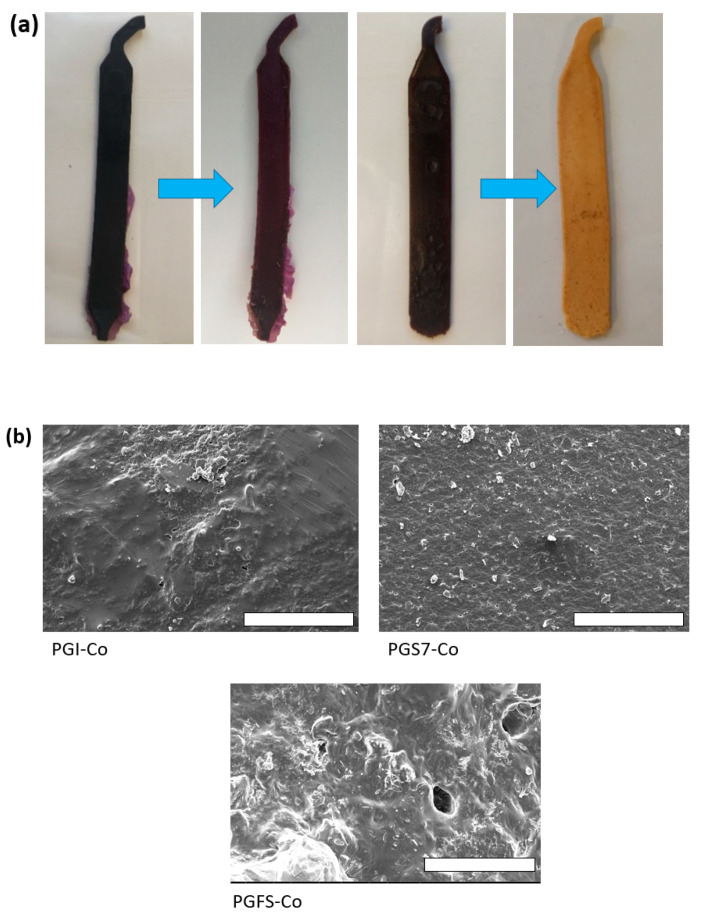
(**a**) Evolution of PGS7-Co and PGI-Co after immersion in a deionized water bath. (**b**) Comparison between infrared spectra of elastomer sheets from PGI-Co, PGS7-Co, and PGFS-Co, along with their classic melt-polycondensation-produced counterparts. (**c**) Photographs of FE-SEM observations of PGI-Co, PGS7-Co, and PGFS-Co surfaces (scale bars = 400 µm). (**d**) Cytocompatibility tests following the ISO10993 standard protocol and performed with PGI-Co, PGFS-Co, PGS7-Co, PGS3, PGS6, and PGS7.

**Figure 11 polymers-14-03829-f011:**
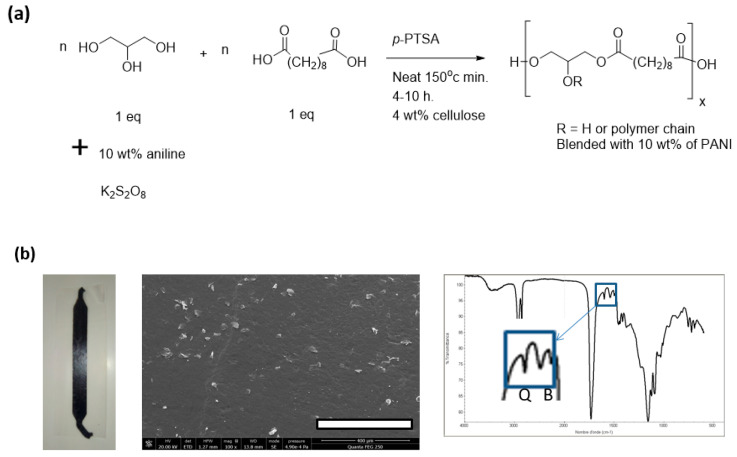
(**a**) Scheme for the production of the PGS8 prepolymer. (**b**) from left to right: image of the bioplastic composite sheet of PGS8; photograph of the FE-SEM observation of the PGS8 surface (scale bar = 400 µm); and its corresponding FTIR spectrum. R—H or polymer chain; Q—Quinoid units; B—Benzoid units.

**Figure 12 polymers-14-03829-f012:**
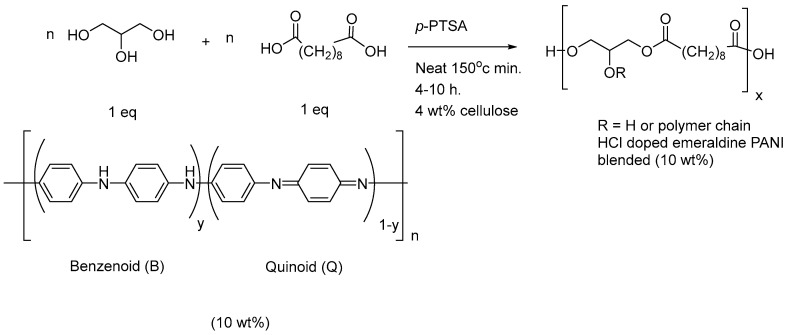
Global procedure for the production of robust PGS9 sheets containing 10 wt% emeraldine PANI.

**Figure 13 polymers-14-03829-f013:**
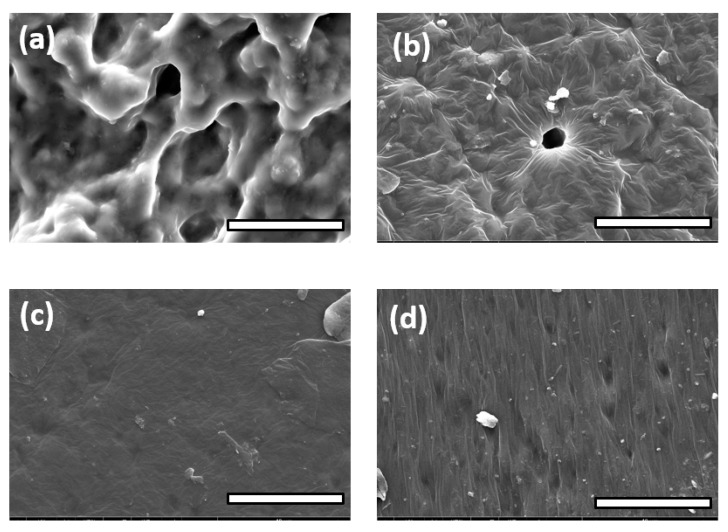
Images of FE-SEM observations from (**a**) PGS7, (**b**) PGS7-Co, (**c**) PGS8, and (**d**) PGS9. Scale bars = 40 µm.

**Figure 14 polymers-14-03829-f014:**
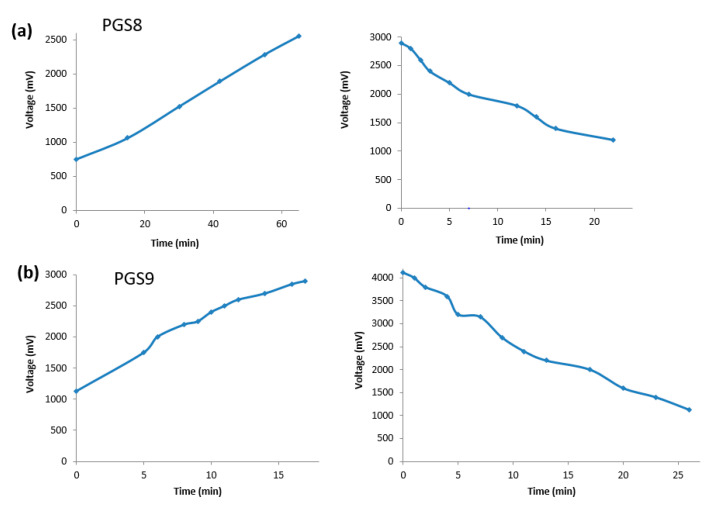
Plots of the dynamic responses of PGS8 and 9 pH sensor candidates versus time and their respective inverse decrease time: charge and discharge for (**a**) PGS8 and (**b**) PGS9. The sheets were powered by a constant current of 20 V.

**Table 1 polymers-14-03829-t001:** List of prepolymers, describing their appearances and modes of production.

Entry	Additive (4 wt%)	Temp (°C)	Original Characteristics
Pre-PGI1	-	160	Not sticky, elastic
Pre-PGI2	-	150	Not sticky, elastic, translucent
Pre-PGS1	-	160	Sticky, soft, translucent
Pre-PGS2	NaCl ^1^	160	Sticky, brown, soft, translucent
Pre-PGS3	Microcrystalline cellulose	150	Not sticky, soft, translucent
Pre-PGS4	Nanocellulose	150	Sticky, soft, translucent
Pre-PGS5	Pluronic^®^ F127	150	Sticky, soft, translucent
Pre-PGS6	PSSNa-PEDOT ^2^	150	Sticky, rigid, dark blue
Pre-PGS7	PANI ^2^	150	Sticky, dark brown
Pre-PGFS1	-	150	Sticky, elastic, white
Pre-PGFS1	NaCl ^1^	150	Sticky, elastic, cream

^1^ NaCl added to the bulk as a porogen agent at 25 wt% concentration. ^2^ Electroconductive polymers synthesized from their respective monomers (EDOT or aniline) in glycerol, in the presence of KPS, prior to polycondensation.

**Table 2 polymers-14-03829-t002:** DSC results of PGI1 and 2, PGFS1 and 2, and PGS1–7.

Event (°C)	PGI1	PGI2	PGFS1	PGFS2	PGS1	PGS2	PGS3	PGS4	PGS5	PGS6	PGS7
Tm1		55			37		16	35	9	6	11
Tm2		83						74	38	36	
Ttc	177	136	190	56	141	124	128	130	183	241	

With Tm1 and Tm2 as melting temperatures; Ttc as temperature of thermal crosslinking.

## Data Availability

Raw data are stored in-house at UTC, and can be made available upon request.

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
