# Peer review of "Solvent-Free Production by Extrusion of Bio-Based Poly(glycerol-co-diacids) Sheets for the Development of Biocompatible and Electroconductive Elastomer Composites"

_polymers, 2022, doi:10.3390/polym14183829_

Round 1

Reviewer 1 Report

The manuscript “ Solvent-free production by extrusion of biobased poly(glycerol-co-diacids) sheets for the development of bio-compatible and electroconductive elastomer composites“ written by Shengzhi Ji et al. was reviewed. In the introduction, the authors of the article made a search from current sources that correspond to the studied issue. They also described the raw materials used and the preparation of polymers in detail. However, in section 2.7 DSC there is written just very short information, and the range is not fit with table 2. Please confirm the information with the results. As regards the results and discussion section, I recommend rewriting and supplementing the results with relevant measurements for the most part, or when it comes to hypotheses, use relevant citations. For example in section 3.2 there is figure 3 with bad resolution/quality and in the text is written something about transparency „PGI1 was almost transparent“ and „The best transparency was observed with…“; how was it measured, do you use some equipment or is it just your subjective view? It should be quantified. In section 3.3 there is written „maybe due to further ester…“ if it is true, use the citation of the article. If not you should measure that. Figure 7, 8 and 10 are with bad quality/resolution. In DSC measurements you set the range 40 – 250 °C, but the results of Tm1 and Tm2 are under 40 °C; how is it possible? Usually, elastomers are amorphous polymers, also in this article is written about transparency, so I expect that in this polymer is not a crystalline phase or yes? Because in the label of table 2 there is written that Tm1 and Tm2 are melting temperatures. In the paragraph below table 2, "PGI2 appeared to be more amorphous“ is written. Please avoid the expression as appear, maybe, seem, too close, the best etc. and use the relevant phrase or citation of articles. In conclusion, is written that the studied elastomer is the best and has a strong potential, it can be, but It should be compared with other materials and research.

The topic of the article is interesting, but the processing of the results and their discussion need to be reworked so that they are clearly understandable to readers and there are only relevant passages and not assumptions.

I recommend publishing this article after incorporating all my comments above.

Author Response

Referee 1

However, in section 2.7 DSC there is written just very short information, and the range is not fit with table 2. Please confirm the information with the results.

In the paragraph 2.7, I corrected the temperature range for all DSC experiments.

As regards the results and discussion section, I recommend rewriting and supplementing the results with relevant measurements for the most part, or when it comes to hypotheses, use relevant citations.

For example in section 3.2 there is figure 3 with bad resolution/quality and in the text is written something about transparency „PGI1 was almost transparent“ and „The best transparency was observed with…“; how was it measured, do you use some equipment or is it just your subjective view? It should be quantified.

We have enlarged the size and the resolution of the Figure 3.

In section 3.3 there is written „maybe due to further ester…“ if it is true, use the citation of the article. If not you should measure that.

In the section 3.3. we have modified the sentences in the best way. We have added a recent reference in the text and the bibliography part about the synthesis of PLA grafted cellulose via melt polycondensation process. We have also changed the numbering in the text.

Figure 7, 8 and 10 are with bad quality/resolution.

We have enlarged the Figure 7 and 8 to improve the quality. I decomposed the Figure 10 into fourth distinct images that display better quality.

In DSC measurements you set the range 40 – 250 °C, but the results of Tm1 and Tm2 are under 40 °C; how is it possible?  

I checked the native datas. There was a mistake in the text. Indeed, we have performed DSC measurements between 0 and 250°C.

Usually, elastomers are amorphous polymers, also in this article is written about transparency, so I expect that in this polymer is not a crystalline phase or yes? Because in the label of table 2 there is written that Tm1 and Tm2 are melting temperatures.

We have already analyzed all polymer sheets by XRD. In fact, they show amorphous phase along the material. In fact, when we have reported phenomena as Tm1 and Tm2, we thought about loss of inner crystallized water departure followed by hydrogen bonding network that soften the elastomer sheets. Then, we have detailed the meaning of each phenomena in the Table 2.

In the paragraph below table 2, "PGI2 appeared to be more amorphous“ is written. Please avoid the expression as appear, maybe, seem, too close, the best etc. and use the relevant phrase or citation of articles.

We have removed these words and replaced them by most suitable expressions everywhere in the text.

Reviewer 2 Report

This manuscript reports the synthesis of poly(glycerol-co-diacids) and their applications as elastomers sheets—it unfortunately contains inconsistency and lacks some basic chemical characterizations. Authors need to conduct NMR and GPC analysis of the prepolymers to show how different additives impact their structures and molecular weights. Especially in the case of itaconic acid/glycerol condensation, authors must check if the polymerization conditions isomerized itaconic acid to citraconic acid.

The manuscript needs extensive editing to improve its clarity. The title is confusing and there are occasional typos/awkward use of languages in the main text. Overall, the manuscript needs additional experiments and extensive revisions.

Here are some major flaws:

Page 3 Section 2.2, paragraph one: The authors seem to use terms like “prepolymer” and “elastomer’ interchangeably. Prepolymer, however, typically means soluble polymeric species prior to thermal crosslinking in the field of poly(diacid-co-glycerol ), while elastomer is acquired after crosslinking prepolymers.

Page 4 Section 2.4: 24-48 hours of incubation in saturated NaHCO3 might cause degradation. The author should study if the washing condition degrades the polymers.

Page 5: Figure 2 is not labeled; A2, R1, and B3 structures are not present.

Page 6, paragraph one, line one: Prepolymers were produced by melt condensation with or without catalysts (p-TSA, Co(Ac)2). GPC and NMR are necessary to see if catalysts made the polymerization more efficient and caused any side reaction.

Page 6, Table 1: Aniline has an amine group—can the amine react with COOH during melt condensation to alter the polymer structures and properties?

Figure 7, Figure 9, Figure 10: Need much better resolutions—figure 10 is illegible.

Page 12, Table 2: Tg was not characterized, and DSC traces should be attached in SI; TGA analysis will also be beneficial.

Author Response

Referee 2:

Here are some major flaws:

Page 3 Section 2.2, paragraph one: The authors seem to use terms like “prepolymer” and “elastomer’ interchangeably. Prepolymer, however, typically means soluble polymeric species prior to thermal crosslinking in the field of poly(diacid-co-glycerol ), while elastomer is acquired after crosslinking prepolymers.

We have modified the paragraph and its title by removing all prepolymer terms that were not appropriate. We have replaced it by elastomer.

Page 4 Section 2.4: 24-48 hours of incubation in saturated NaHCO3 might cause degradation. The author should study if the washing condition degrades the polymers.

Except for PGI that was subjected to some kind of degradation after immersion in the NaHCO3 bath, that was outside of our expectation, we have never observed degradation by naked eyes for other elastomer sheets. The SEM pictures were done after treating as described in the experimental section. In the paragraph 2.4. we added the comment about this step realized prior to the other analysis and characterizations such as FE-SEM observation.

Page 5: Figure 2 is not labeled; A2, R1, and B3 structures are not present.

We have modified the Figure 2 by identifying the nature of R1 and R in the image.

Page 6, paragraph one, line one: Prepolymers were produced by melt condensation with or without catalysts (p-TSA, Co(Ac)2). GPC and NMR are necessary to see if catalysts made the polymerization more efficient and caused any side reaction.

It is difficult to evaluate by NMR or SEC of the incidence of the catalyst on the polymerization level due to the infusible character of our elastomer sheets. The only one possible side-reaction should be the formation of macrocycle of elastomer, but the phenomenon is in direct competition with the ramification of the system.

Page 6, Table 1: Aniline has an amine group—can the amine react with COOH during melt condensation to alter the polymer structures and properties?

The aniline was not employed during the elastomer composite production. Only PANI is employed for producing the PGS-PANI sheets by extrusion. As the best of my knowledge, the terminal primary amine of PANI is not enough reactive to form amide function from an ester function of the polyester in this condition. We have never observed the appearance of both signals of amide functions for PGS7 or PGS8 on FT-IR spectra.

Figure 7, Figure 9, Figure 10: Need much better resolutions—figure 10 is illegible.

We have enlarged the Figure 7 and 8 to improve the quality. I decomposed the Figure 10 into fourth distinct images that display better quality.

Page 12, Table 2: Tg was not characterized, and DSC traces should be attached in SI; TGA analysis will also be beneficial.

Tg as referred as Tm1 and Tm2. Unfortunately we have never conducted any TGA experiments for our samples because our objective is to work in range of temperature where living organisms could evolve. We added the DSC thermograms of every sample reported in the Table 2 in the ESI.

Round 2

Reviewer 1 Report

The revised manuscript “ Solvent-free production by extrusion of biobased poly(glycerol-co-diacids) sheets for the development of bio-compatible and electroconductive elastomer composites“ written by Shengzhi Ji et al. was reviewed. The article was revised according to the reviewer's comments, but there are still passages that need to be revised or explained.

I recommend the publication of this work after correcting these findings:

In line 59 – „CaTiO3 ceramic [2635]“ Check the citation link.

In line 169 – „15 Kv accelerating“ – 15 kV

In line 242 – „synthesis The synthesis of“ - Check the sentence.

Table 1 last row – there is written the same entry „Pre-PGFS1“ as before row – check it.

Figure 3 – the scale of 8 cm is upside down, and the quality of the pictures is not sufficient to assess the transparency of the samples. Furthermore, the methodology for assessing the transparency of the samples was not presented.

In lines 300-301 – there is written „The best transparency was observed with Pluronic ® F127, for PGS2 and PGFS2.“ Check this sentence if fits with Figure 3 and Table1.

Author Response

Response to reviewer 2 (Round 2)

I recommend the publication of this work after correcting these findings:

In line 59 – „CaTiO3 ceramic [2635]“ Check the citation link.

We have corrected the wrong numbering and replaced it by [26].

In line 169 – „15 Kv accelerating“ – 15 kV

We corrected the unit to kV.

In line 242 – „synthesis The synthesis of“ - Check the sentence.

We modified the text as follow: 3…synthesis. The synthesis of…”

Table 1 last row – there is written the same entry „Pre-PGFS1“ as before row – check it.

We modified the Table 1. The second row is different, here we added NaCl to a comparable sample of Pre-PGFS1.

Figure 3 – the scale of 8 cm is upside down, and the quality of the pictures is not sufficient to assess the transparency of the samples. Furthermore, the methodology for assessing the transparency of the samples was not presented.

 After discussing this issue with all the authors, we have reconsidered the transparency of our material and replaced everywhere in the manuscript the word “transparent” by translucent. (and more especially in the paragraph starting from line 330 just after the Table 1) In fact, our PGS-based sheets were slightly opaque in some cases; even if we have reached a level near to transparency for PGS4.

In lines 300-301 – there is written „The best transparency was observed with Pluronic ® F127, for PGS2 and PGFS2.“ Check this sentence if fits with Figure 3 and Table1.

We have modified the sentence as follow: “Some kind of translucency was observed with Pluronic ® F127, for PGS2 and PGFS2. The »

Reviewer 2 Report

The authors addressed all my concerns. 

Round 3

Reviewer 1 Report

Thanks to the authors for clarifying my comments. I now recommend this manuscript for publication.